# Phage Lytic Protein CHAPSH3b Encapsulated in Niosomes and Gelatine Films

**DOI:** 10.3390/microorganisms12010119

**Published:** 2024-01-06

**Authors:** Verdiana Marchianò, Ana Catarina Duarte, Seila Agún, Susana Luque, Ismael Marcet, Lucía Fernández, María Matos, Mª del Carmen Blanco, Pilar García, Gemma Gutiérrez

**Affiliations:** 1Department of Physical and Analytical Chemistry, University of Oviedo, Julián Clavería 8, 33006 Oviedo, Spaincblanco@uniovi.es (M.d.C.B.); 2Department of Chemical and Environmental Engineering, University of Oviedo, Julián Clavería 8, 33006 Oviedo, Spain; sluque@uniovi.es (S.L.); marcetismael@uniovi.es (I.M.); matosmaria@uniovi.es (M.M.); 3Instituto de Productos Lácteos de Asturias (IPLA-CSIC), Paseo Río Linares s/n., 33300 Villaviciosa, Spain; catarina.leal@ipla.csic.es (A.C.D.); seila.agun@ipla.csic.es (S.A.); lucia.fernandez@ipla.csic.es (L.F.); 4DairySafe Group, Instituto de Investigación Sanitaria del Principado de Asturias (ISPA), 33011 Oviedo, Spain; 5Instituto Universitario de Biotecnología de Asturias, University of Oviedo, 33006 Oviedo, Spain

**Keywords:** antimicrobial activity, niosomes, endolysin, encapsulation, gelatine films

## Abstract

Antimicrobial resistance (AMR) has emerged as a global health challenge, sparking worldwide interest in exploring the antimicrobial potential of natural compounds as an alternative to conventional antibiotics. In recent years, one area of focus has been the utilization of bacteriophages and their derivative proteins. Specifically, phage lytic proteins, or endolysins, are specialized enzymes that induce bacterial cell lysis and can be efficiently produced and purified following overexpression in bacteria. Nonetheless, a significant limitation of these proteins is their vulnerability to certain environmental conditions, which may impair their effectiveness. Encapsulating endolysins in vesicles could mitigate this issue by providing added protection to the proteins, enabling controlled release, and enhancing their stability, particularly at temperatures around 4 °C. In this work, the chimeric lytic protein CHAPSH3b was encapsulated within non-ionic surfactant-based vesicles (niosomes) created using the thin film hydrating method (TFH). These protein-loaded niosomes were then characterized, revealing sizes in the range of 30–80 nm, zeta potentials between 30 and 50 mV, and an encapsulation efficiency (EE) of 50–60%. Additionally, with the objective of exploring their potential application in the food industry, these endolysin-loaded niosomes were incorporated into gelatine films. This was carried out to evaluate their stability and antimicrobial efficacy against *Staphylococcus aureus*.

## 1. Introduction

The widespread use of antibiotics across different sectors, especially throughout the last century, has led to a rapid increase in the overall resistance of microbial populations. The direct consequence of this phenomenon has been a notable loss in the efficacy of antibiotherapy in the clinic, leading to increased morbidity and mortality. Therefore, antibiotic resistance represents a public health threat with significant social and economic costs. In this context, the World Health Organization (WHO) has published a list of bacteria for which new antibiotics are urgently needed, which includes methicillin- and vancomycin-resistant strains of *Staphylococcus aureus* (MRSA and VRSA, respectively) [1]. In 2020, the prevalence of MRSA was 25% or higher in 10 out of 40 surveyed countries/areas [2]. Furthermore, there is a rising trend in the isolation of MRSA strains in farm animals. This trend is alarming, as animals can acquire and disseminate antibiotic-resistant strains along the food chain.

*S. aureus* is commonly found on the skin and mucosa of different animals, including humans. However, this Gram-positive bacterium is frequently armed with a wide array of virulence factors, including multiple toxins and host immune system evasion factors [3]. This arsenal makes *S. aureus* an important human pathogen responsible for a variety of infections, especially those associated with the hospital environment. Additionally, this species is a major causative agent of food-borne diseases in humans due to the production of heat-resistant enterotoxins. To make matters worse, this bacterium can readily form biofilms on both inert surfaces and living tissues. Biofilm formation is a complex process which involves the initial adhesion of bacterial cells to a surface, followed by the production of an extracellular matrix. This matrix primarily consists of a combination of exopolysaccharides, proteins, and DNA. Importantly, biofilm cells exhibit a greater ability to resist antibiotics and disinfectants compared to planktonic cells. Indeed, the prevention and destruction of biofilms remain challenging tasks. These will require the development and subsequent implementation of new strategies [4].

Bacteriophages (phages) are viruses that infect and kill bacteria. As a result, they constitute a promising alternative or complementary strategy to the use of antibiotics and disinfectants [5]. Moreover, phage-derived lytic proteins also exhibit interesting antimicrobial properties that can help to fight against antibiotic-resistant and biofilm-forming bacteria. Typically, phages degrade the structural peptidoglycan present in the bacterial cell wall using two classes of lytic proteins: virion-associated peptidoglycan hydrolases (VAPGHs), which degrade peptidoglycan in the initial steps of the infection, and endolysins, which help to release the phage progeny during the late phase of the lytic cycle. These enzymes can be used as antimicrobial agents by targeting bacteria from the outside, accessing the peptidoglycan, and destroying the cell walls, ultimately leading to cell lysis. Lytic proteins are relatively easy to produce, safe for humans and the environment, target-specific, and do not easily select for resistant mutants [6]. Furthermore, the modular structure of lytic proteins active against Gram-positive bacteria facilitates the design of chimeric proteins via domain shuffling. This approach frequently leads to the identification of new variants displaying improved lytic activity. However, in order to implement the use of lytic proteins in clinical or food industry applications, it is essential to ensure their stability, enabling them to successfully reach their target [7].

One option for the stabilization of lytic proteins is their encapsulation in vesicles, a method commonly used for drug delivery [8]. Vesicles are carriers formed by an aqueous core surrounded by a lipid layer (membrane) in which it is possible to encapsulate both hydrophilic and hydrophobic molecules. In the case of drugs, encapsulation is known to enhance their pharmacodynamic properties as well as to reduce some potential side effects [9]. Other advantages of vesicles include their high biocompatibility, physical and chemical stability, good affinity towards drugs, and easy route of administration.

Different types of vesicles can be identified based on their main membrane components, with liposomes and niosomes being the most common. The vesicle membranes of liposomes are primarily composed of phospholipids, whereas niosomes utilize non-ionic surfactants [10].

Niosomes have several advantages over liposomes, including higher stability, easier access to raw materials, and a broader variety of available materials. Additionally, these materials exhibit lower toxicity and high compatibility with biological systems, and offer flexibility for surface modification [11].

Studies exploring the encapsulation of phage lytic proteins have confirmed the viability and efficacy of this method [12]. Moreover, in the case of Gram-negative bacteria, liposome-mediated endolysin encapsulation systems allow these proteins to penetrate the outer membrane and reach the peptidoglycan without the need for a membrane permeabilizer [13]. Nonetheless, it is still necessary to build upon the success of these preliminary studies. Testing additional encapsulation strategies will help to pinpoint the most adequate vesicle composition for each specific application. It is crucial to consider that colloidal systems containing lytic proteins should ideally have average sizes below 80–130 nm to effectively penetrate bacterial biofilm channels [14].

However, the encapsulation of lytic proteins in niosomes has been less explored compared to other vesicle types. Given the versatility of niosomes in terms of membrane composition, their use for encapsulating these proteins could offer several advantages. This includes the potential to produce positively charged niosomes, a characteristic that numerous studies have shown to positively impact biofilm elimination [15]. In this study, niosomes were formulated using the surfactant Span 60 and Cholesterol, both recognized for their suitability in encapsulating various compounds [11]. Additionally, the cationic surfactant cetyltrimethylammonium bromide (CTAB) was incorporated to enhance the positive charge of the niosome membrane and to leverage its known antimicrobial activity [16].

Besides vesicle encapsulation, endolysins can also be incorporated into different materials such as gelatine or starch to manufacture antimicrobial films. In the case of the food industry, for instance, these proteins could be incorporated into food packaging matrices. This could address the challenge concerning pathogens that survive traditional food processing methods [17]. The development of active food packaging based on endolysins would represent an innovative step in enhancing food safety, adding a crucial layer of protection against both spoilage and pathogenic microorganisms.

Previous studies have already shown the antimicrobial and antibiofilm activity of various staphylococcal phage lytic proteins [6]. Indeed, some proteins, such as LysH5 and CHAPSH3b, have proven to be effective for both biofilm removal and inhibition of biofilm formation [18]. Based on the existing data, we selected the chimeric protein CHAPSH3b, which was obtained by fusion of the CHAP domain from HydH5 (encoded by the *S. aureus* phage vB_SauS-phiIPLA88) and the SH3b cell wall binding domain from lysostaphin [19]. This protein demonstrated antistaphylococcal activity in growth medium and milk, and other interesting parameters for its future application as an antimicrobial [20].

Taking all of the above into account, this work had two main objectives. First, we sought to assess the antibiofilm potential of the antistaphylococcal lytic protein CHAPSH3b encapsulated in niosomes. The prepared niosomes were characterized in terms of size, zeta potential, encapsulation efficiency (EE), and antibiofilm activity. Subsequently, we investigated the development of a gelatin-based film incorporating the encapsulated lytic protein and explored its antimicrobial properties. The aim of using both free niosomes and gelatin films was to expand the applications of CHAPSH3b towards both clinical and food settings, producing final products with antimicrobial activity and non-human toxicity [21].

## 2. Materials and Methods

### 2.1. Chemical Compounds, Bacterial Strains, and Proteins

The surfactants used as components of the niosomes membrane included Span^®^ 60 (Sigma-Aldrich, Saint Louis, MI, USA), cholesterol stabilized at 96% (Acros Organics, Newark, NJ, USA), and CTAB (Sigma-Aldrich). All of these components were dissolved in absolute ethanol (J.T. Baker, Avantor, Allentown, PA, USA). For the aqueous phase, phosphate buffer saline (PBS) was prepared using tablets (Oxoid, Hampshire, UK) according to the manufacturer’s instructions. The PBS composition was 137 mM NaCl, 2.7 mM KCI, 8 mM Na_2_HPO_4_, and 2 mM KH_2_PO_4_, with a pH of 7.4. Gelatine films were prepared using gelatine from porcine skin (Sigma-Aldrich ref. G1890) and glycerol (Sigma-Aldrich 99.5% ref. G7893).

The lytic protein CHAPSH3b was overexpressed and purified following previously described methods [22]. The quantity and quality of the purified enzyme were determined by the Quick Start™ Bradford Protein Assay Kit (Bio-Rad, Madrid, Spain) and SDS-PAGE visualization, respectively. The bacterial strains used in this study were *S. aureus* 15981 [23] and *S. aureus* Sa9 [24]. They were routinely grown at 37 °C in tryptic soy broth (TSB) (Scharlau Microbiology, Barcelona, Spain) with shaking, or on plates containing TSB supplemented with 2% (*w*/*v*) agar (Roko, S.A., Llanera, Spain) (TSA) or Baird–Parker agar plates (AppliChem, Darmstadt, Germany).

### 2.2. Preparation and Characterization of Niosomes

#### 2.2.1. Preparation of Niosomes

Preliminary experiments were conducted to determine the most appropriate combination of the non-ionic surfactant Span 60, cholesterol, and CTAB. Different ratios of Span 60 and CTAB were tested until the desired vesicle size was registered (30–80 nm) to ensure the desired size for efficient biofilm penetration.

The synthesis of niosomes was carried out using the thin film hydration (TFH) method [16], with some modifications. The synthesis involved dissolving the vesicle membrane components (total membrane compounds: 0.25 g, with a mass ratio 1:1:0.5 of Span 60, CTAB, and cholesterol) in an organic phase (10 mL of ethanol). This was followed by the vacuum evaporation of the solvent using a rotary evaporator. To achieve a homogeneous, transparent, and dry film, temperature was controlled at 40 °C using a thermostatic water bath, pressure at 70 mbar, and rotation speed at 135 rpm. Initially, the lytic protein was dissolved in 25 mL of the aqueous phase at a concentration of 8 μM. This solution was then used to hydrate the thin film in the flask. Two different solutions were tested as the aqueous phase: pure Milli-Q water and PBS. Next, the mixture was incubated at 45 °C in a water bath (Figure 1), deliberately avoiding higher temperatures to prevent lytic protein destabilization. Finally, the prepared vesicles underwent sonication to reduce their size using a Branson Ultrasonics Sonifier SFX150 (Nuevo Laredo, Tamaulipas, Mexico) for 10 min at an amplitude of 45%, with 500 W power and a frequency of 20 kHz. To produce empty niosomes, the same procedure was followed, excluding the incorporation of the lytic protein.

#### 2.2.2. Size and Morphology Characterization

The size and zeta potential of the niosomes were measured using dynamic light scattering (DLS) on a Zetasizer NanoZS series (Malvern Instruments Ltd., Malvern, UK), taking non-diluted aliquots of 100 µL for size determination and 2 mL for zeta potential determination. Both measurements were made at 25 °C [16,25].

Morphology was assessed by transmission electron microscopy (TEM) with a JEOL-2000 Ex II transmission electron microscope (Tokyo, Japan). For TEM analysis, a sample drop was placed on a carbon-coated copper grid. Excess sample was then removed with filter paper. Subsequently, a drop of 2% (*w*/*v*) phosphotungstic acid solution (PTA) was applied to the grid and left for 1 min. After removing excess staining agent with filter paper, the sample was air-dried. The stained and fixed niosomes were then ready for observation under the microscope.

#### 2.2.3. Niosome Purification

Purification was carried out to remove non-encapsulated lytic protein by ultracentrifugation at 35,000 rpm for 50 min at 4 °C (using an OptimaTM MAX ultracentrifuge at 130,000 RPM, Beckman Coulter, Brea, CA, USA). The supernatant was then filtered through polyethersulfone (PES) syringe filters of 0.22 µm pore diameter (TermoFischer, Waltham, MA, USA) for subsequent analysis by high-performance size exclusion chromatography (HPSEC).

#### 2.2.4. Determination of the Protein Encapsulation Efficiency (EE)

To evaluate the encapsulation efficiency, two mL of vesicle suspension in either pure Milli-Q water or PBS was treated with 25% ethanol. This was performed to break the membrane bilayer and release the encapsulated endolysin [25]. Following this treatment, samples were analyzed using HPSEC chromatography. Measurement of the peak area allowed for protein quantification. Sample separation was performed at room temperature with an HPLC 1100 series chromatograph (Agilent Technologies, Santa Clara, CA, USA) equipped with a Superose™ 12 10/300 GL column (Cytiva, Buckinghamshire, UK) by injecting between 5 and 100 µL per sample. The eluent was 0.5 M NaCl and 0.2 M Tris(hydroxymethyl)aminomethane (Sigma-Aldrich), prepared in a solution with ultrapure water, buffered at pH 8 using HCl, and filtered through a 0.2 μm nylon disc filter. The eluent flow rate was 0.5 mL/min, and a diode array detector was used to record absorbance at 224, 234, 244, 254, and 280 nm. Calibration was carried out using proteins of known molecular weight, and the exclusion limits were determined with Blue Dextran 2000 (Cytiva, Buckinghamshire, UK) and acetone (Panreac, Barcelona, Spain), respectively.

An ESI-Q/TOF-type mass spectrometer (IMPACT-II, Bruker, Billerica, MA, USA) coupled to a UHPLC liquid chromatograph (DIONEX ULTIMATE3000, Thermo Scientific, Waltham, MA, USA) was employed to determine the molecular weight of the proteins eluted in the previously described HPSEC chromatography. A BIOshellTM A400 Protein C4 column (150 × 2.1 mm, 3.4 μm, GE HealthCare, Chicago, IL, USA) and the SUPELCOR LC Program (with the column oven at 70 degrees) were used. The mobile phases consisted of 0.1% formic acid in water (A) and 0.1% formic acid in acetonitrile (B). The gradient program used is indicated in Table 1. EE was calculated according to Equation (1), utilizing the areas measured from the chromatograms of purified systems (without free lytic protein) and non-purified niosomes (containing encapsulated and free lytic protein).
(1)$$EE\left(\%\right)=\frac{\left(CHAPSH3b\;in\;purified\;niosomes\right)}{\left(CHAPSH3b\;in\;non-purified\;niosomes\right)}\times 100\;$$

#### 2.2.5. Antimicrobial Activity of Niosomes

For the antimicrobial tests, 25 mL samples containing both empty and protein-loaded niosomes were ultracentrifuged at 35,000 rpm for 50 min at 4 °C (Optica^TM^ MAX ultracentruge, Beckman Coulter). The supernatant containing free non-encapsulated protein was removed, and the pellet was resuspended with 250 µL of PBS or Milli-Q water in order to remove surfactants, empty niosomes, and contaminants from the suspension [26].

An overnight culture of *S. aureus* 15981 was diluted (1:100 *v*/*v*) in fresh TSB medium supplemented with 0.25% (*v*/*v*) glucose (Merk, Darmstadt, Germany) (TSBg). Then, 1 mL aliquots were inoculated into each well of a 24-well polystyrene microtiter plate (Thermo Scientific, Nunclon Delta Surface). These plates were incubated for 24 h at 37 °C to allow for biofilm development. Afterwards, the planktonic phase was removed, and the biofilms were washed twice with PBS prior to treatment with 0.5 mL of PBS (control), empty niosomes, or protein-loaded niosomes. These treatments were applied for durations of 1, 2, 4, 6, and 24 h. After incubation, the planktonic phase was removed, and the adhered biofilms were washed twice with PBS. To assess the efficacies of the different treatments, the number of viable attached cells was quantified using the spot test. Biofilms were first scraped from the bottom of the well and resuspended in PBS. Afterwards, 10 µL droplets from tenfold serial dilutions of these cell suspensions were spotted onto TSA plates and allowed to dry. These plates were then incubated at 37 °C for 24 h. All experiments were performed in triplicate. The cell counts obtained from these experiments were used to calculate the number of colony-forming units (CFU/cm^2^) per area unit using Equation (2):(2)$$\frac{CFU}{cm^{2}}=Volume\;of\;the\;suspension\times \frac{CFU}{Volume\;inoculated\;on\;the\;plate}\times \frac{\left(Dilution\;Factor\right)}{Area}$$

### 2.3. Synthesis and Characterization of Gelatine Films

Gelatin films were prepared by directly mixing 0.7 g of porcine skin gelatin with 0.245 g of glycerol in 10 mL of water-based suspensions containing either empty or protein-loaded niosomes. The resulting mixtures were then placed in a water bath at 40 °C for 25–30 min to dissolve the gelatine. Once dissolved, the solutions were poured into Petri dishes and allowed to dry in an oven at 35 °C for 24 h. After drying, the films were carefully peeled off from the dishes using tweezers. Four types of gelatin films were prepared: (i) plain gelatin films as a control, (ii) gelatin films containing free lytic protein, (iii) films with empty niosomes, and (iv) films with protein-loaded niosomes. All endolysin-containing films had protein concentrations of 0.057 µM/g of film.

#### 2.3.1. Films Characterization

The morphologies of the different films, including both surface and cross-sections, were observed using a scanning electron microscope (SEM) (JSM-5600, JEOL, Peabody, MA, USA), and measurements were carried out at 20 kV with a working distance of 10–20 mm. Prior to observation, films were cut into squares measuring 0.2 × 0.2 mm. These samples were then placed on stubs and coated with gold using a Polaron SC7620 instrument at 20 mA, with a time coating of 120 s.

#### 2.3.2. Films’ Antimicrobial Activity

The antimicrobial activity of the films was evaluated using *S. aureus* Sa9. The concentration of protein added to the gelatin was consistently 8 μM, resulting in a final assay concentration of 2 μM. Each film was divided into 4 identical pieces. These pieces were then immersed in 10 mL of TSB medium with an initial concentration of 10^7^ CFU/mL of *S. aureus* Sa9 and incubated for 4 h or 24 h at 37 °C with shaking. After incubation, the number of viable cells was determined by serial dilution and plating on Baird–Parker medium, and they were then incubated at 37 °C for 16 h. Additionally, the films were stored for 14 days at 4 °C and subsequently retested for their residual antimicrobial activity. Data from at least three independent biological replicates were analyzed using a two-tailed Student’s *t*-test. *p*-values less than 0.05 were considered statistically significant.

### 2.4. Statistical Analysis of Data

All experiments were conducted in triplicate, and the results are presented as arithmetic mean values with standard deviations. To analyze differences between test groups, an analysis of variance (ANOVA) was conducted using IBM^®^ SPSS^®^ Statistics 25. Significant differences between groups were determined using Fischer’s least significant difference (LSD) test. The biofilm data were statistically analyzed using a two-way ANOVA, followed by Dunnett’s multiple comparisons test, employing GraphPad Prism 6 software. In all statistical tests, *p*-values less than 0.05 were considered to indicate significant differences.

## 3. Results

### 3.1. Protein Encapsulation and Characterization of the Prepared Niosomes

Previous studies have demonstrated the promising antistaphylococcal activity of the lytic protein CHAPSH3b, as well as its efficacy as an antibiofilm agent [18,27]. Here, we aimed to explore whether these properties could be further improved through increasing protein stability by nanoencapsulation. More specifically, CHAPSH3b was encapsulated in niosomes that contained the non-ionic surfactant Span 60 (Sp60) as a main compound, cholesterol (Cho) as a stabilizer, and CTAB as a positively-charged co-stabilizer. The most appropriate molar ratio between the vesicle components was found to be 2:1:2 (Sp60:Cho:CTAB). The total concentration of membrane compounds in the organic phase was 15 g/L. The concentration of the purified protein used for encapsulation was 1.6–2.9 mg/mL (total yield was 6.4–11.6 mg/L of *E. coli* culture). This stock was then diluted at 8 µM (243.2 μg/mL) in pure (Milli-Q) water or phosphate buffer saline (PBS) prior to encapsulation. This concentration was selected based on prior findings, which indicated its effectiveness for antibiofilm activity [27]. In parallel, empty niosomes were obtained and used as negative controls in all assays.

The size and surface charge of the resulting niosomes, with and without protein, are summarized in Table 2, and the particle size distribution is presented in Figure 2. As can be observed in Figure 2 and Table 2, empty niosomes prepared in ultrapure water had sizes of 100 nm and highly positive surface charges (55.2 mV). This can be attributed to the presence of the cationic surfactant CTAB. In contrast, niosomes containing the protein CHAPSH3b were smaller, approximately half the size of the empty niosomes, and exhibited a narrower particle size distribution according to dynamic light scattering (DLS) analysis. However, the zeta potential of the loaded niosomes was similar to that of the empty niosomes (46 ± 5 mV). Interestingly, the average vesicle size doubled when using PBS buffer as aqueous hydration medium instead of water. Nevertheless, the size of the niosomes prepared in PBS also decreased after loading the protein, mirroring the pattern observed with water-based niosomes. In general, the presence of the protein did not significantly alter the positive charge of the niosomes. A slight reduction in charge was observed when using Milli-Q water in the aqueous phase. This reduction was likely due to interactions between the protein and the CTAB molecules at the vesicle membrane surface (Figure 2 and Table 2).

The morphological analysis of the empty and protein-loaded niosomes by transmission electron microscopy (TEM), presented in Figure 3, confirmed the DLS data. This confirmed that all niosomes exhibited a spherical shape with a mean size of around 50–200 nm. It was also observed that niosomes in PBS showed larger sizes and a slight greater tendency to agglomerate than those suspended in water. However, it is important to take into account that no large differences in agglomeration were expected, since all zeta potential values fall around 28–55 mV, indicating values large enough to avoid high agglomeration without resisting high repulsion in any case.

### 3.2. Protein Encapsulation Efficiency (EE) Determined by Size Exclusion Chromatography

The concentration of encapsulated proteins was determined by size exclusion chromatography and quantifying the absorbance at a wavelength of 280 nm, which is commonly used for protein detection. Prior to assessing the EE, CHAPSH3b protein solutions prepared in Milli-Q water and PBS (2 to 12 µM) were tested by size exclusion chromatography (Figure 4). The column’s void volume resulted in an exclusion time of 14.5 min, measured with Blue Dextran 2000. The smallest molecule used for column calibration, acetone, had an elution time of 54 min. The protein solution contained a significant amount of imidazole (MW 68 g/mol), which appeared at 55 min. In this medium, solute–agarose interactions occur despite the ionic strength used in the mobile phase. There are solute–agarose interactions that result in delayed elution times with respect to the proteins and peptides used for column calibration (BSA and its dimer, carbonic anhydrase, alfa albumin, citochrome C and aprotinin). To confirm the correct allocation of the protein, given the numerous peaks found, fractions of the eluate were collected between 26 and 55 min and further analyzed by UHPLC-ESI-Q/TOF MS (ultrahigh pressure liquid chromatography coupled to quadrupole-time-of-flight mass spectrometry with electrospray ionization). The protein was identified in the peak corresponding to an elution time of 44.6 min, with a possible aggregate eluting at 42.5 min. The rest of the UV-absorbing matter (other proteins, bacterial culture debris, and small molecules) present in the protein solution also showed different behavior depending on the medium. PBS’s ionic strength seemed to enhance protein stabilization in solution, which resulted in higher absorbance values (please notice that the scale was 4x in the case of PBS). To some extent, PBS also shielded the protein from aggregation, as there were fewer peaks visible at earlier elution times. The number of peaks found after the exclusion time was also significantly larger in ultrapure water, indicating that more molecules were interacting with the column matrix, thereby delaying their elution.

The presence and relative concentration of CHAPSH3b in the niosomes were then analyzed using the same method. In this case, however, we tested a wider range of wavelengths (224 to 280 nm), as some vesicle components also absorb UV light. Protein absorbance showed a maximum peak at 254 nm, while the peak for imidazole became predominant at lower wavelengths (Figure 5). The protein was easily identified and quantified at 280 and 254 nm (Figure 5). Moreover, analysis of the empty niosomes and solvent revealed that there was no overlapping peak at the protein elution time (44.6 min). Additionally, as can be seen in Figure 6, the peaks corresponding to the protein could be easily identified in mixtures containing the protein and niosomes together.

To assess the protein encapsulation efficiency (EE), the encapsulated CHAPSH3b was released from the loaded niosomes. It was then separated from the membrane compounds by ultracentrifugation. Aliquots from the supernatant of the broken niosomes containing the released protein, and from the overall mixture (without supernatant separation), were analyzed by HPSEC (Figure 5). The EE was estimated from the relative absorbance of the peak observed between 44 and 45 min, which is marked with a green arrow in Figure 5. The results indicate that ultrapure water yields more encapsulated protein according to the chromatography peak dimensions (EE = 62%) compared to PBS (EE = 51%). However, the use of PBS yielded better protein stability, evidenced by a more distinct main peak in the chromatogram.

### 3.3. Antimicrobial Activity of CHAPSH3b-Loaded Niosomes

Based on these results, we chose to evaluate the antimicrobial activity of niosomes prepared using PBS as a solvent against staphylococcal biofilms. *S. aureus* 15981 was chosen as a target strain for these tests because it develops strong, polysaccharide-rich biofilms [28]. Table 3 shows that there was a 4.68-log reduction in treated biofilm cells compared with control biofilms after the first hour of treatment when using CHAPSH3b-containing niosomes. The highest reduction in viable cells occurred after 4 h of incubation. At that point, the number of viable cells in the samples treated with protein-loaded niosomes was below the detection level. Empty niosomes also displayed antimicrobial activity and curtailed growth thanks to the presence of CTAB, an ammonium salt with known antibacterial properties. However, bacterial removal derived from this treatment was always below that obtained with protein-loaded niosomes. It is worth mentioning that biofilms treated with free protein also showed lower viable cell counts than the control. Nevertheless, this reduction was never as notable as that obtained using niosomes. The efficacy of the different treatments remained clear after 6 h of incubation. However, regrowth of the bacterial population was observed after 24 h of incubation, probably indicating the loss of antimicrobial activity of the protein.

### 3.4. Characterization of Gelatine Films Containing Encapsulated or Free CHAPSH3b

With the aim of broadening the potential applications of encapsulated CHAPSH3b in the food industry, we prepared pork skin gelatin films containing free protein, empty niosomes, or protein-loaded niosomes. Morphological characterization of these films by scanning electron microscopy (SEM) revealed differences between gelatine films prepared with PBS alone (control films) (Figure 7A) and those films containing empty niosomes (Figure 7B), free protein (Figure 7C), and encapsulated protein (Figure 7D). Films containing only PBS exhibited a homogeneous appearance. In contrast, films containing empty niosomes presented a rougher appearance. This is likely indicative of the inclusion of the colloidal niosomes on the gelatine matrix (Figure 7B). Moreover, those films with free (Figure 7C) or encapsulated protein (Figure 7D) exhibited dots that may indicate the presence of protein agglomerates inside the gelatine matrix films, especially in the case of films containing free protein. This suggests that protein agglomeration may occur within the films. This was particularly noticeable in films containing free protein.

### 3.5. Antimicrobial Activity of CHAPSH3b-Containing Gelatine Films

After examining the structures of the various films, we evaluated the antimicrobial activity of gelatin films containing only PBS, empty niosomes, protein-loaded niosomes, and free protein against a suspension of *S. aureus* Sa9 cells in growth medium. This strain has been consistently used in studies by our team on the efficacy of staphylococcal endolysins against planktonic cells. Films containing free CHAPSH3b reduced, but did not completely prevent, bacterial growth (Table 4). This result demonstrated that the protein remained stable and active following the film preparation process. However, the empty niosomes displayed significant antimicrobial activity, which resulted in inhibition of bacterial growth below the detection level (Table 4). This significant activity of the empty niosomes masked the effect of the encapsulated protein, thereby hindering our ability to observe any potential synergistic or additive interactions between the niosomes and the enzyme.

Besides assessing the efficacy of the prepared films immediately after preparation, some films were stored for 14 days to evaluate their long-term stability and efficacy. Again, we observed that the film-embedded free protein exhibited lower antibacterial activity. In contrast, films containing encapsulated protein or empty niosomes demonstrated complete inhibition of bacterial growth (Table 4).

## 4. Discussion

Encapsulating phage lytic proteins could be an effective strategy to enhance their stability and antimicrobial potential. Indeed, several recent studies have highlighted the benefits of encapsulating endolysins in niosomes [13] or solid nanoparticles [22]. Furthermore, recent studies have demonstrated the effectiveness of this approach in therapeutic applications [12,29]. However, to date, there are no specific studies addressing the use of encapsulated endolysins in the food industry.

In this work, the antistaphylococcal lytic protein CHAPSH3b was successfully encapsulated in non-ionic niosomes by combining Span60, cholesterol, and CTAB at concentrations that ensured the desired vesicle size. Niosomes were prepared at 45 °C, a temperature at which the protein remains stable and retains its activity [30]. The presence of the cationic surfactant CTAB confers the niosomes membrane with a positive charge. This phenomenon is less noticeable when encapsulation is performed in PBS instead of Milli-Q water, as the salts shield the positive charge of CTAB due to ionic interactions [31]. Nonetheless, the overall charge is still positive enough to ensure colloidal stability [32], and also to exert the antibacterial activity commonly associated with this compound [33]. Indeed, the estimated charge of the vesicle surface (25–55 mV) aligns with findings from other studies that involve encapsulation with cationic surfactants [34].

Regarding size, the protein-loaded niosomes exhibited a two-fold decrease (from 100–200 nm to 40–80 nm) when compared to the empty niosomes. In contrast, previous studies have shown that encapsulated biocompounds generally increase in vesicle size [28,35]. The observed behavior may indicate a tendency of lytic proteins to locate at the vesicle membrane. They likely act as costabilizers, reducing the critical packing parameter of the self-assembling surfactant. Interestingly, surfactants with a high hydrophilic–lipophilic balance exhibit similar behavior when added to the vesicle membrane [36]. It is possible that the effect on vesicle size may correlate with the molecular weight of the encapsulated biocompound, with a more noticeable vesicle size increase when encapsulating high-molecular-weight biocompounds [26]. Other studies have demonstrated that the presence of certain additives, such as some costabilizers, leads to the formation of much larger niosomes. This is notably the case with glycerine and polyethylene glycol, commonly used to increase the EE [36]. A similar effect on the final vesicle size has been observed when using lipophilic costabilizers like cholesterol [37].

In this study, we observed a higher encapsulation efficiency (EE) for niosomes formed with MilliQ water as the hydration medium compared to those using PBS, correlating with smaller niosome sizes. This outcome suggests a propensity for lytic proteins to be encapsulated within the niosome membrane rather than in the aqueous core. It has been generally observed that the encapsulation of compounds, which tend to localize at the membrane layer, is more efficient with the thin-film hydration (TFH) method. This contrasts with other techniques such as ethanol injection or microfluidics, where unilamellar vesicles are predominantly formed [38].

Analysis of protein encapsulation by size exclusion chromatography showed that the use of PBS as an aqueous hydration medium during vesicle formulation results in better protein stability. This was evidenced by a more defined main peak around 45 min in PBS-based samples compared to water-based samples, as shown in Figure 5. Therefore, given that stabilization is our main objective, PBS was the chosen solvent in the aqueous phase for the encapsulation of CHAPSH3b in subsequent experiments.

CHAPSH3b has already been demonstrated as an effective antibiofilm agent [18,27]. In this study, we show that the encapsulation of this protein leads to even better results, although this can be partly attributed to the antimicrobial activity of the niosomes themselves. Indeed, the cationic surfactant CTAB is an ammonium salt with known antimicrobial activity that provokes cell lysis [39]. Similarly to our findings, empty liposomes (DMPC:DOPE:CHEMS, molar ratio 4:4:2), designed to encapsulate lysins Pa7 and Pa119, also exhibited a lytic effect against *Pseudomonas aeruginosa* cultures, likely due to membrane destabilization [40]. Despite the impact of CTAB on cell viability, it is important to note that the highest antimicrobial efficacy was observed when testing endolysin-loaded niosomes. This indicates that encapsulation may enhance the protein’s stability and/or facilitate a synergistic interaction between the lytic protein and the vesicle components. Elucidating the mechanism behind this interaction warrants further investigation.

Moreover, the use of CTAB confers the prepared niosomes with a positive charge, which is known to have beneficial effects on biofilm penetration [41]. An important aspect to consider is that biofilms have microchannels that allow the entry of water and exchanges with the external environment. These channels, approximately 200 µm in diameter, allow the entry of small-sized niosomes and enable the delivery of antimicrobials to target bacteria within the biofilm [42]. Furthermore, various studies have shown that positively charged niosomes could electrostatically interact with bacterial cells, which have negatively charged surfaces, thereby facilitating their entrance. The best range of zeta potential to achieve this effect is around 40–50 mV [43].

It is worth mentioning that only a few staphylococcal endolysins have been successfully encapsulated to date. For instance, LysRODI, encoded by the phage vB_SauM_phiIPLA-RODI, was encapsulated in pH-sensitive liposomes and subsequently tested against *S. aureus* biofilms [44]. The results revealed a significant reduction in viable cells after 24 h of incubation when treated with the liposome-encapsulated protein. This result suggests that encapsulation of these antimicrobials opens new possibilities for their delivery. Another example is LysMR-5, an endolysin derived from phage MR-5, which was encapsulated in alginate–chitosan nanoparticles. It did not exhibit any loss of structural integrity or bioactivity after entrapment [45]. Moreover, there are notable examples of endolysins targeting other Gram-positive bacteria. For instance, Vázquez et al. [43] successfully encapsulated the lytic protein Cpl-711 (ChiDENPs-711) in chitosan nanoparticles. This enhanced its stability and enabled the release of over 90% of the active enzybiotic within approximately 2 h [46].

Gelatine is a natural biopolymer known for its biodegradability, biocompatibility, affordability, and ease of sourcing. In the food industry, gelatine is also used in the form of films to protect food against oxidation and microbial contamination, allowing for its long-term preservation [47]. Here, we tested the possibility of embedding CHAPSH3b into biodegradable gelatine films suitable for food-packaging purposes, as well as clinical or pharmaceutical applications. Food packaging is an area of growing interest due to its high impact on food product quality; over time, edible films have become widely used. If these films can serve as vehicles for transporting bioactive compounds, their applicability can be extended even further. Our results prove that CHAPSH3b retains its antimicrobial activity after the film preparation procedure, which involves several temperature changes. This finding is crucial, as it demonstrates the potential of endolysins for the development of new antimicrobial materials that may help to fight the antibiotic resistance crisis. However, when added to gelatin films, the empty niosomes exhibited very high antimicrobial activity. This, unfortunately, masked the activity of the encapsulated lytic protein. Indeed, no bacterial growth was detected even after 14 days of film storage when using empty niosomes or encapsulated protein. Therefore, further studies are needed to determine whether vesicle encapsulation prior to film preparation can enhance protein stability. This would involve testing a range of lytic proteins and various experimental conditions. To the best of our knowledge, this is the first time that a lytic protein has been embedded in this type of film. In contrast, bacteriophages have already been successfully tested as part of biodegradable films [48] and gelatine films [49]. For example, a previous work demonstrated the enhanced activity of phage PBSE191 against *Salmonella* when embedded in a polyvinyl alcohol film compared to the free phage [50]. Also, it must be emphasized that, as far as we are aware, no studies have explored the activity of phages or endolysins within a film matrix after storage.

## 5. Conclusions

The present study demonstrates the feasibility of encapsulating endolysins with antimicrobial activity against *S. aureus* in non-ionic niosomes. Antibiofilm tests indicate that niosomes loaded with endolysins can display higher antimicrobial activity compared to free endolysins or empty niosomes. We also proves the feasibility of preparing gelatin films containing endolysin, either free or encapsulated in niosomes, which had high inhibition activity against planktonic bacteria even after 14 days of storage. A potential additive or even synergistic effect between CTAB and phage protein is behind this effect; therefore, future studies should aim to quantify the extent to which protein encapsulation enhances endolysin activity versus the antimicrobial effect of the niosomes. Additionally, it will be necessary to examine parameters concerning the behavior of endolysin-loaded niosomes in foods, such as release dynamics and shelf-life stability.

## Figures and Tables

**Figure 1 microorganisms-12-00119-f001:**
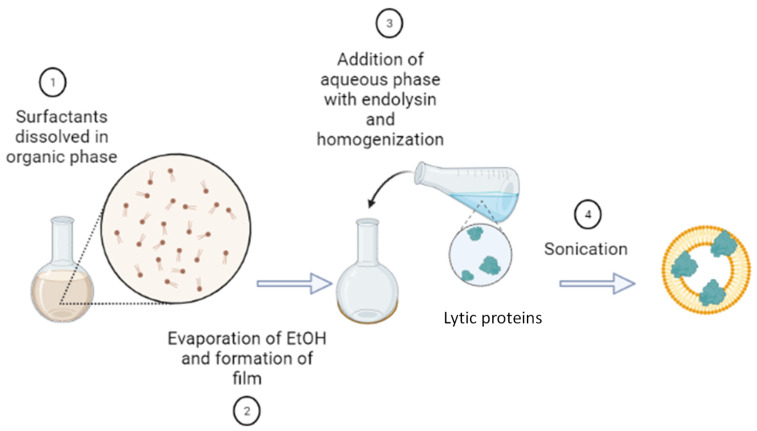
Thin film hydration method used for the synthesis of niosomes containing the lytic protein CHAPSH3b.

**Figure 2 microorganisms-12-00119-f002:**
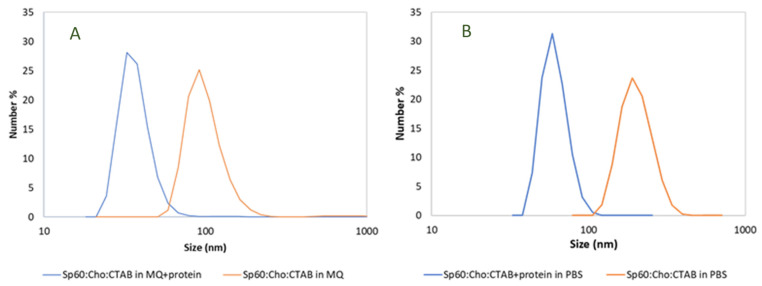
Particle size distribution of niosomes prepared in (**A**) pure Milli-Q water and (**B**) PBS obtained by DLS.

**Figure 3 microorganisms-12-00119-f003:**
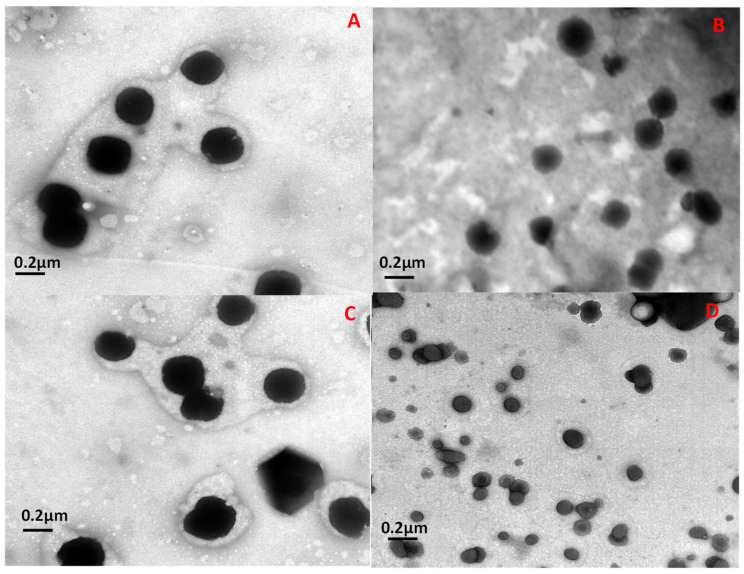
TEM images of niosomes stained with 2% (*w*/*v*) phosphotungstic acid (PTA). (**A**) Empty niosomes in water; (**B**) protein-loaded niosomes in water; (**C**) empty niosomes in PBS; (**D**) protein-loaded niosomes in PBS.

**Figure 4 microorganisms-12-00119-f004:**
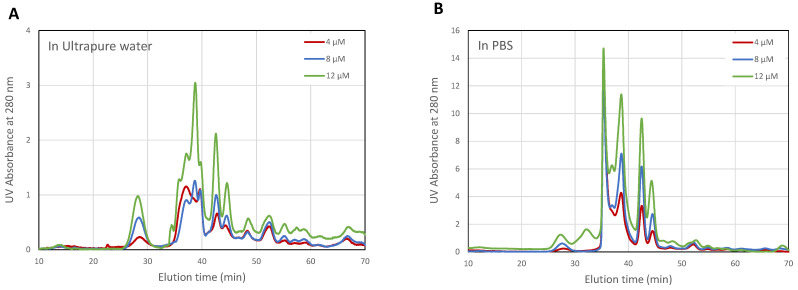
Protein elution profiles obtained after measurement of the absorbance at 280 nm (A_280_) of fractions eluted by exclusion chromatography of CHAPSH3b: 4, 8, and 12 µM in ultrapure water (**A**) and in PBS (**B**).

**Figure 5 microorganisms-12-00119-f005:**
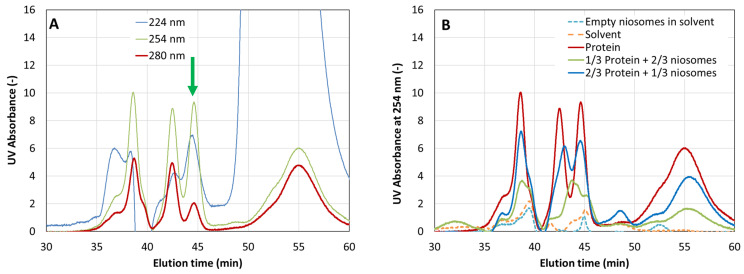
Elution profiles of fractions obtained by molecular exclusion chromatography of the solvent (ethanol), empty niosomes, and protein/niosomes/solvent after measurement at (**A**) different wavelengths (blue line: 224, green line: 254, and red line: 280 nm), and (**B**) different fractions at 254 nm. Red line: protein CHAPSH3b, green line: mixture of empty niosomes (1/3) and protein (2/3) solution; blue line: mixture of empty niosomes (2/3) and protein (1/3) solution. The green arrow indicates the CHAPSH3b fraction.

**Figure 6 microorganisms-12-00119-f006:**
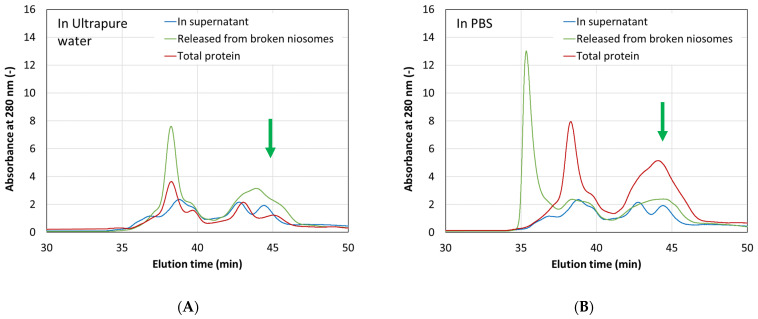
HPSEC chromatographs (at 280 nm) for different samples eluted in ultrapure water (**A**) and in PBS (**B**). Protein-loaded niosomes were first separated from the supernatant (blue line), and then the niosomes were broken (green line) to release CHAPSH3b. A mixture of supernatant and broken niosomes (total protein) was also analyzed (red line).

**Figure 7 microorganisms-12-00119-f007:**
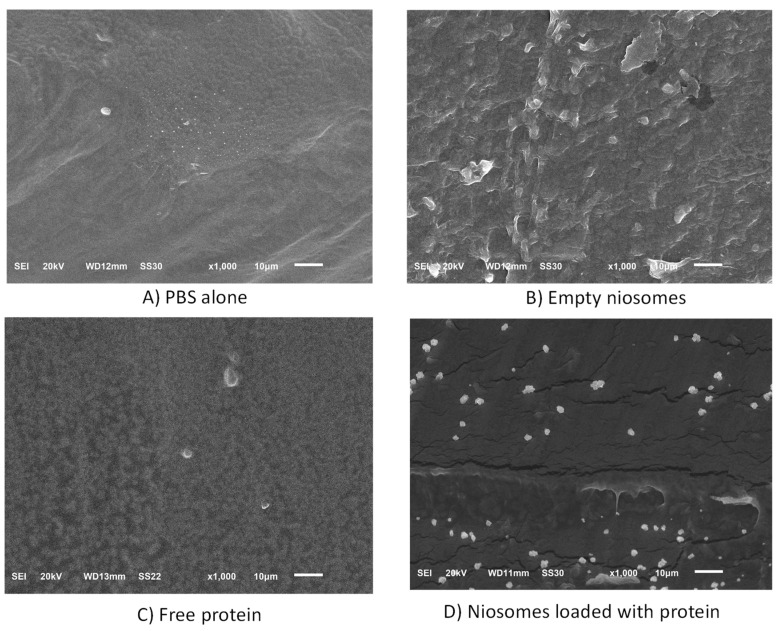
Analysis by SEM of gelatine films obtained with (**A**) PBS alone, (**B**) empty niosomes, (**C**) free protein, and (**D**) niosomes loaded with protein.

**Table 1 microorganisms-12-00119-t001:** UHPLC (ultra-high-performance liquid chromatography) gradient program.

**Time (min)**	**Gradient (Percentage of Solvent B by Volume)**
0	5
15	90
17	90
17.1	5
21	5

**Table 2 microorganisms-12-00119-t002:** Mean particle size and zeta potential of niosomes prepared in different aqueous media.

Formulation	Aqueous Phase	Size (nm)	Zeta Potential (mV)
Sp60: Cho: CTAB (no protein)	Milli-Q water	100 ± 27	55 ± 2
Sp60: Cho: CTAB + CHAPSH3b (8 µM)	Milli-Q water	38 ± 18	46 ± 5
Sp60: Cho: CTAB (no protein)	PBS buffer	205 ± 46	28 ± 2
Sp60: Cho: CTAB + CHAPSH3b (8 µM)	PBS buffer	77 ± 21	30 ± 4

**Table 3 microorganisms-12-00119-t003:** 24-h-old biofilms of strain *S. aureus* 15981 treated with free CHAPSH3b, empty niosomes, or protein-loaded niosomes and compared to an untreated control.

	Values Correspond to Log (CFU/cm^2^)			
Time (h)	PBS Buffer	CHAPSH3b (8 µM)	Empty Niosomes	CHAPSH3b (8 µM) Loaded Niosomes
1	8.42 ± 0.06	7.85 ± 0.02	5.27 ± 0.90	3.74 ± 1.71 **
2	8.24 ± 0.13	7.28 ± 0.60	4.13 ± 3.60 *	3.90 ± 3.40 **
4	7.62 ± 0.71	6.50 ± 0.46	1.33 ± 2.30 ****	0.00 ± 0.00 ****
6	8.72 ± 0.58	7.32 ± 0.06	1.15 ± 1.99 ****	0.00 ± 0.00 ****
24	6.81 ± 0.30	6.36 ± 0.68	3.22 ± 2.79 *	3.07 ± 2.66 *

Values correspond to log (CFU/cm^2^). Data represent the means ± standard deviations of three independent experiments. Values with asterisks are statistically different from the untreated control at each time point according to two-way ANOVA using Dunnett’s multiple comparisons test. * *p*-value < 0.05, ** *p*-value < 0.01 and **** *p*-value < 0.0001.

**Table 4 microorganisms-12-00119-t004:** Antimicrobial activity of gelatine films against a bacterial suspension in growth medium. *S. aureus* Sa9 cultures were exposed to gelatine films containing free CHAPSH3b protein, empty niosomes, or encapsulated protein and compared to an untreated control. All samples were incubated for 4 or 24 h at 37 °C with shaking.

	Values Correspond to Log (CFU/mL)			
Incubation Time (h)	PBS	Free CHAPSH3b (8 μM)	CHAPSH3b (8 µM) Loaded Niosomes	Empty Niosomes
**After preparation**				
**4**	8.6 ± 0.30	5.18 ± 0.17 *	0.00 ± 0.00 *	0.00 ± 0.00 *
**24**	9.50 ± 0.17	7.43 ± 0.32 *	0.00 ± 0.00 *	0.00 ± 0.00 *
**After 14 days of storage**				
**4**	9.02 ± 0.12	8.57 ± 0.22	0.00 ± 0.00 *	0.00 ± 0.00 *

* *p*-value < 0.05.

## Data Availability

Data are contained within the article.

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
