# Peer review of "Phage Lytic Protein CHAPSH3b Encapsulated in Niosomes and Gelatine Films"

_microorganisms, 2024, doi:10.3390/microorganisms12010119_

Round 1

Reviewer 1 Report

Comments and Suggestions for Authors

Interesting and necessary for minor revision before acceptance

1, Methods 2.2.1, 2.2.2, 2.2.3,  2.2.4, 2.3.1 and 2.3.2, these sections,  any key reference should be cited here if possible?

2, Line 171, RP-HLPC any or which model? 

3, Figure 3, better resolution with high clearness?

4, Figure 4, μmol marked in Figure is not correct?

5, Table 3, h, shopuld be placed with Time together in Top line? And Control should be directly noted as what Buffer or water?and "Values correspond to Log (CFU/cm2)" are adivsed to move into Top line with index together in bracket as ( ) if possible? And similar modification in Table 4?

6, Line 655, 658, and others, Staphylococcus Aureus should be typed as in italics? other similar errors? and other format style errors as Caapital and small letters shouldm be united in your text? 

7,  3.1. Protein encapsulation and characterization of the prepared vesicles

Any or wjhich is the best Harvest ratio and optimization for your end product or preparation?

8, And any cost factor or analysis was considered? in Discussion or other section? 

9, And release dynamics or ratio of target antimicrobial agent during the assured period of shelf-life related to field of food and other industry ? more in details?

10, Any simple safety test for your end product necessary in your hands by youself or out sourcing if possible ?

(END, 28DEC2023, JW)

Author Response

Interesting and necessary for minor revision before acceptance

  1. Methods 2.2.1, 2.2.2, 2.2.3, 2.2.4, 2.3.1 and 2.3.2, these sections, any key reference should be cited here if possible?

References related on the synthesis of vesicles and gelatine films had been included in the revised version of the manuscript.

  1. Line 171, RP-HLPC any or which model? 

HPSEC was utilized instead of RP-HPLC, and this has been duly corrected in the text.

  1. Figure 3, better resolution with high clearness?

Quality of Figure 3 has been improved in the revised version of the manuscript.

  1. Figure 4, marked in Figure is not correct?

Legends have been modified to denote properly the concentration.

  1. Table 3, h, should be placed with Time together in Top line? And Control should be directly noted as what Buffer μmol or wate?and "Values correspond to Log (CFU/cm2)" are adivsed to move into Top line with index together in bracket as ( ) if possible? And similar modification in Table 4?

Table 3 and 4 have been improved and modified according to the referee’s indication.

  1. Line 655, 658, and others, Staphylococcus Aureus should be typed as in italics? other similar errors? and other format style errors as Capital and small letters should be united in your text? 

Manuscript has been revised. Italic letters have been used for Staphylococcus aureus throughout the revised version of the manuscript.

7,3.1. Protein encapsulation and characterization of the prepared vesicles

Any or which is the best Harvest ratio and optimization for your end product or preparation?

The harvested product consists of films embedded with lytic proteins, which retain their antimicrobial properties. This production yields sustainable and safe materials suitable for use in food packaging applications.

8, And any cost factor or analysis was considered? in Discussion or other section? 

The final protein concentration achieved after overexpression in E. coli ranged between 1.6 and 2.9 mg/ml, corresponding to a total yield of 6.4 to 11.6 mg/L of E. coli culture. This information has now been indicated in the text.

As for the production costs, making an accurate prediction at this stage is challenging. Further research, particularly in developing efficient upscaling strategies, is necessary before beginning to estimate such costs. Currently, our laboratory's research is focused on testing various protocols to upscale endolysin production for successful industrial-level manufacturing. However, these studies are still in their early stages and are too preliminary to be included in this manuscript.

9, And release dynamics or ratio of target antimicrobial agent during the assured period of shelf-life related to field of food and other industry? more in details?

Similar to the upscaling of protein production, the release dynamics and stability of encapsulated endolysin in food products are subjects that currently interest our team, and further research in these areas is planned for the coming years. However, specific details are not yet available for inclusion in this manuscript. We have highlighted the necessity for information on these properties in the manuscript's conclusion section.

10, Any simple safety test for your end product necessary in your hands by yourself or out sourcing if possible?

Existing literature provides information on the non-toxicity of gelatin films and lytic proteins, supporting the suitability of incorporating lytic protein into gelatin films for use in final food packaging applications (Tuyuftin et al, 2021; Nandi et al., 2022).

References:

Andrey A. Tyuftin, Joe P. Kerr,  Food Packaging and Shelf Life, 29, 2021, 100688

Front. Med., 04 November 2022, doi: https://doi.org/10.3389/fmed.2022.1047752

This information has been included in the introduction section of the manuscript.

Reviewer 2 Report

Comments and Suggestions for Authors

The manuscript entitled "Phage lytic protein CHAPSH3b encapsulated in niosomes and gelatine films" is an original study that provides interesting findings on the possibility to encapsulate proteins in niosomes and further incorporate them into a film that can be used as a food packaging. This strategy could be applied for the prevention of MRSA contamination of food and have a positive impact on the antibiotic therapy in general. The manuscript is overall well written and structured. However, there are still some issues that would require clarification in order for the article to be suitable for publishing in Microorganisms. Below, is the list of my comments and recommendations:

1. The Introduction section would benefit from addition of references to prove some statements.

2. The vesicles are very generally mentioned in the Introduction and the statement between lines 84-87 is too general and not supported by literature data. Especially the statement "Studies exploring the encapsulation of phage lytic proteins have confirmed the viability and efficacy of this method" - which are these studies? what is "this method" referred to?

3. There are numerous types of vesicles- liposomes, ethosomes, niosomes, transferosomes, emulsomes, and the generalization in the present study is therefore not appropriate. Better focus on niosomes and the choice of excipients for their preparation is highly advised, especially regarding the Span 60 over other types of Span and CTAB which has its own antibacterial properties.

4. The authors prepare and discuss only niosomes and the titles is also referred to this specific type of nanovesicles. Therefore, in the manuscript text niosomes should be discussed and not referred to as nanovesicles in the methods, results and discussion.

5. In the Introduction section terms such as "other interesting parameters" , "incorporated into different materials" are too general and should be rephrased and clarified.

6. In my opinion, the nanovesicles and the gelatin film are not "synthesized" rather prepared and the corresponding sections should be rephrased.

7. The authors state several times "desired vesicle size (30-80 nm)". How was this size chosen as desired? Please, elaborate.

8. The niosomes preparation procedure is not clearly explained. The transition temperature of Span 60 according to literature is 53 0C, why the authors chosen to work at 450C? What is the purpose of the procedure: "the mixture was incubated ... in a water bath"? Figure 1 shows sonication step which is not given in the explanation of the procedure. How exactly the niosomes have been prepared?

9. DLS procedure is not properly defined - "taking aliquots of 100 μl and 2 mL, respectively" refers to what? Where the dispersions diluted? What was the scattering angel and temperature of the measurement?

10. Equation 1 shows how the encapsulation efficiency was calculated. What does "purified vesicles" mean? 

11. SEM conditions are not provided, such as acceleration, sputter coater?

12. What does the following statement mean "the protein ... was purified at 1.6-2.9 mg/ml" (line 271)?

13. The authors state that there is possible interaction between protein and CTAB molecules. Transform infrared spectroscopy could provide information on the speculation.

14. The TEM images (Figure 3) have different scale bars and it is confusing for the comparison. The charge of the vesicles doesn't change that much to be the explanation for agglomeration of the niosomes in the image 3D. Please, provide better images or elaborate on the observed differences.

15. Figure 6 - "a mixture of broken vesicles" mixture with what? The caption of the figure is not clear - eluted in ultrapure water and in PBS (if I understood correctly the mobile phase contains NaCl, Tris(hydroxymethyl)aminomethane in ultrapure water with pH 8 or is is water and PBS?) Why the total protein is almost double in PBS?

16. The antimicrobial activity of the empty niosomes is significantly higher than the lytic protein on its own. I am not convinced that the loaded niosomes have better antimicrobial activity than the empty ones. How could this result contribute to the application of CHAPSH3b?

17. In the results section the authors discuss "long term stability" after 14 days storage without mentioning the storage conditions. This duration cannot be referred to as long-term stability. Stability testing should be added as a method and properly explained.

18. The authors state that the EE in MilliQ water is higher than the one in PBS, yet the niosomes are smaller. The current explanation of the change in the size is not sufficient. The different EE is not addressed in the discussion section. T he literature example show the reason for larger vesicles and not smaller which is the case in the current study.

19. Abbreviations should be introduced the first time being mentioned (e.g. PBS, PES, PVA, etc.)

Comments on the Quality of English Language

Minor English editing is required due to some typos-, equation 1,  line 338 UV light and some others.

Author Response

The manuscript entitled "Phage lytic protein CHAPSH3b encapsulated in niosomes and gelatine films" is an original study that provides interesting findings on the possibility to encapsulate proteins in niosomes and further incorporate them into a film that can be used as a food packaging. This strategy could be applied for the prevention of MRSA contamination of food and have a positive impact on the antibiotic therapy in general. The manuscript is overall well written and structured. However, there are still some issues that would require clarification in order for the article to be suitable for publishing in Microorganisms. Below, is the list of my comments and recommendations:

  1. The Introduction section would benefit from addition of references to prove some statements.

Additional references have been incorporated into the introduction section of the revised manuscript to substantiate all statements made.

  1. The vesicles are very generally mentioned in the Introduction and the statement between lines 84-87 is too general and not supported by literature data. Especially the statement "Studies exploring the encapsulation of phage lytic proteins have confirmed the viability and efficacy of this method" - which are these studies? what is "this method" referred to?

In previous studies, lytic proteins have been encapsulated within various types of vesicles, such as liposomes and niosomes, achieving encapsulation efficiency values in the range of 60-80% (Viruses 2018, 10(9), 495).

Additionally, their encapsulation in other colloidal systems like emulsions, and the application of techniques such as spray drying or freeze drying, have been explored as well (Advances in Colloid and Interface Science, 2017, 249, 100-133).

This information has been comprehensively included in the revised version of the manuscript.

  1. There are numerous types of vesicles- liposomes, ethosomes, niosomes, transferosomes, emulsomes, and the generalization in the present study is therefore not appropriate. Better focus on niosomes and the choice of excipients for their preparation is highly advised, especially regarding the Span 60 over other types of Span and CTAB which has its own antibacterial properties.

In the introduction section of the revised manuscript, various types of vesicles are described. Additionally, the choice of Span 60, cholesterol, and CTAB as membrane components for the niosomes has been thoroughly justified.

To summarize briefly, Span 60 and cholesterol were chosen for their well-established ability to form stable vesicles, which are effective in encapsulating a diverse range of biocompounds. Furthermore, CTAB was selected for its antimicrobial properties and its capacity to be seamlessly incorporated into the vesicle membrane, thereby imparting a positive charge.

  1. The authors prepare and discuss only niosomes and the titles is also referred to this specific type of nanovesicles. Therefore, in the manuscript text niosomes should be discussed and not referred to as nanovesicles in the methods, results and discussion.

The authors appreciate the suggestion and have consistently referred to 'niosomes' throughout the revised manuscript.

  1. In the Introduction section terms such as "other interesting parameters", "incorporated into different materials" are too general and should be rephrased and clarified.

Sentences have been rephrased in the revised version of the manuscript.

  1. In my opinion, the nanovesicles and the gelatin film are not "synthesized" rather prepared and the corresponding sections should be rephrased.

Sections have been rephrased in the revised version of the manuscript.

  1. The authors state several times "desired vesicle size (30-80 nm)". How was this size chosen as desired? Please, elaborate.

Previous studies in the literature have highlighted the ability of colloidal systems sized between 80-130 nm to penetrate bacterial biofilm channels (Critical Reviews in Microbiology, 2022, 48, 283-302). This information has been incorporated into the revised version of the manuscript..

  1. The niosomes preparation procedure is not clearly explained. The transition temperature of Span 60 according to literature is 53 0C, why the authors chosen to work at 45ºC? What is the purpose of the procedure: "the mixture was incubated ... in a water bath"? Figure 1 shows sonication step which is not given in the explanation of the procedure. How exactly the niosomes have been prepared?

The preparation of niosomes is described with greater detail in the revised version of the manuscript. This procedure has been adapted from previous methods, specifically modified to avoid high temperatures and thereby preserve the properties of the lytic proteins.

  1. DLS procedure is not properly defined - "taking aliquots of 100 μl and 2 mL, respectively" refers to what? Where the dispersions diluted? What was the scattering angel and temperature of the measurement?

A more detailed description of the methodology is included in the revised version of the manuscript. Briefly, the samples were not diluted. For size distribution analysis, aliquots of 100 µL were utilized, and for zeta potential determination, 2 mL of the sample was used. All measurements were recorded at a temperature of 25°C.

  1. Equation 1 shows how the encapsulation efficiency was calculated. What does "purified vesicles" mean? 

The term 'purified vesicles' is used to describe vesicles that have been pelleted and separated from the supernatant containing free lytic protein. This purification step effectively removes all the free lytic protein. This detail has been further clarified in the revised version of the manuscript.

  1. SEM conditions are not provided, such as acceleration, sputter coater?

The revised version of the manuscript now includes detailed information on the SEM (Scanning Electron Microscope) measurements. The instrument used was a JSM-5600 (JEOL, Peabody, MA, USA), and the measurements were conducted at 20 kV with a working distance of 10-20 mm. The morphology of various films, including both surface and cross-sections, was examined. For the observation, films were first cut into 0.2x0.2 mm squares. These samples were then mounted on stubs and subjected to gold coating using a Polaron SC7620 instrument at 20 mA for a coating time of 120 seconds.

  1. What does the following statement mean "the protein ... was purified at 1.6-2.9 mg/ml" (line 271)?

This value represents the concentration of the purified protein, as now specified in the text.

  1. The authors state that there is possible interaction between protein and CTAB molecules. Transform infrared spectroscopy could provide information on the speculation.

The authors acknowledge that this technique has not been employed in the current manuscript. However, they extend their gratitude for the referee's suggestion and plan to incorporate this characterization in future studies.

  1. The TEM images (Figure 3) have different scale bars and it is confusing for the comparison. The charge of the vesicles doesn't change that much to be the explanation for agglomeration of the niosomes in the image 3D. Please, provide better images or elaborate on the observed differences.

In the revised version of the manuscript, all TEM images have been standardized to the same scale bar. Moreover, the selected images more accurately represent the behaviour of the samples. Notably, high agglomeration was not observed in any of the cases. This phenomenon has been discussed in the results section of the revised manuscript.

  1. Figure 6 - "a mixture of broken vesicles" mixture with what? The caption of the figure is not clear - eluted in ultrapure water and in PBS (if I understood correctly the mobile phase contains NaCl, Tris(hydroxymethyl)aminomethane in ultrapure water with pH 8 or is is water and PBS?) Why the total protein is almost double in PBS?

In all instances, the mobile phase for the chromatography analysis remained consistent. However, the purified protein was dissolved either in MilliQ water or in a PBS solution (prepared using water). We observed that the chromatographic signal varied depending on the aqueous phase containing the protein, a variation likely linked to protein stabilization. This observation accounts for the differences noted between the samples in MilliQ water and those in PBS. The term mixture has also been clarified as it corresponds to the combined supernatant + broken niosomes.

  1. The antimicrobial activity of the empty niosomes is significantly higher than the lytic protein on its own. I am not convinced that the loaded niosomes have better antimicrobial activity than the empty ones. How could this result contribute to the application of CHAPSH3b?

Results suggest that empty niosomes exhibit significant antimicrobial activity, likely attributable to the presence of CTAB in the vesicle membrane. Although this activity partially obscures the effect of the endolysin, it undeniably represents a valuable characteristic in the development of antimicrobial products, aiding in the reduction of bacterial load. Furthermore, as demonstrated in Table 3, niosomes loaded with protein display greater efficacy against biofilms compared to their empty counterparts.

  1. In the results section the authors discuss "long term stability" after 14 days storage without mentioning the storage conditions. This duration cannot be referred to as long-term stability. Stability testing should be added as a method and properly explained.

The films were stored at 4°C for 14 days, after which their residual antimicrobial activity was retested. Details of this procedure have been included in both the Materials and Methods section and the abstract of the manuscript's revised version.

  1. The authors state that the EE in MilliQ water is higher than the one in PBS, yet the niosomes are smaller. The current explanation of the change in the size is not sufficient. The different EE is not addressed in the discussion section. The literature example show the reason for larger vesicles and not smaller which is the case in the current study.

For hydrophilic compounds, encapsulation efficiency (EE) tends to be higher in larger vesicles, where a correspondingly larger aqueous core is expected (Colloids and Surfaces B: Biointerfaces, 186, a10711, 2020). In contrast, for lipophilic compounds, where encapsulation occurs in the niosomes' membrane layer, EE is influenced more by the chosen preparation method than by the niosome size (Membranes, 11, 95, 2023; Chemical Engineering Research and Design, 162, 162-171, 2020). This difference is attributed to the morphology of the niosomes. Methods that tend to form multilamellar vesicles, such as the thin hydration method, are particularly effective in achieving higher EE for hydrophobic compounds.

This information has been incorporated into the discussion section of the manuscript's revised version.

  1. Abbreviations should be introduced the first time being mentioned (e.g. PBS, PES, PVA, etc.)

Abbreviations are defined and detailed in the revised version of the manuscript.

Reviewer 3 Report

Comments and Suggestions for Authors

The research proposed by the authors is interesting, especially in the context of finding alternatives to combat the resistance of different types of infectious agents, even in the case of the most virulent, Staphylococcus aureus.

The evaluation of the article was based on:

1.     In the Introduction the authors point out what constitutes an alarm signal in terms of the excessive use of antibiotics, which means the establishment of resistance; the data on Staphylococcus aureus, this Gram-positive bacterium with a particular virulence, are systematized; data on the role of bacteriophages and their derived proteins, those lytic proteins that induce bacterial cell lysis; strategies applied to combat resistance to biofilm formation; the purpose and objectives of the research;

2.     Materials and working methods applied in the research I consider to be properly presented; an interesting sub-chapter is the one related to the synthesis and characterization of nanovesicles that include endolyses; aspects related to the morphological characterization of these nanovesicles, as well as the determination of their size, purification, determination of the protein encapsulation efficiency, the antimicrobial activity of the nanoformulations, the synthesis and characterization of gelatine films, as well as their antimicrobial activities;

3.     The results are described for each sub-stage of the research; the authors express the results in tables, graphs, supporting photomicrographs; all the obtained results are processed statistically;

4.     In the Discussions chapter, the authors correlate the results obtained with the data from the specialized literature and point out the advantages of nanoencapsulations in combating the resistance of different types of pathogens from a therapeutic point of view, as well as from a food point of view; finding strategies that increase the preservation of food sources but without harmfulness;

5.     The conclusions are consistent with the research objectives and open up new research opportunities;

6.     The bibliography is justifiable.

Author Response

The research proposed by the authors is interesting, especially in the context of finding alternatives to combat the resistance of different types of infectious agents, even in the case of the most virulent, Staphylococcus aureus.

The evaluation of the article was based on:

  1. In the Introduction the authors point out what constitutes an alarm signal in terms of the excessive use of antibiotics, which means the establishment of resistance; the data on Staphylococcus aureus, this Gram-positive bacterium with a particular virulence, are systematized; data on the role of bacteriophages and their derived proteins, those lytic proteins that induce bacterial cell lysis; strategies applied to combat resistance to biofilm formation; the purpose and objectives of the research;
  2. Materials and working methods applied in the research I consider to be properly presented; an interesting sub-chapter is the one related to the synthesis and characterization of nanovesicles that include endolyses; aspects related to the morphological characterization of these nanovesicles, as well as the determination of their size, purification, determination of the protein encapsulation efficiency, the antimicrobial activity of the nanoformulations, the synthesis and characterization of gelatine films, as well as their antimicrobial activities;
  3. The results are described for each sub-stage of the research; the authors express the results in tables, graphs, supporting photomicrographs; all the obtained results are processed statistically;
  4. In the Discussions chapter, the authors correlate the results obtained with the data from the specialized literature and point out the advantages of nanoencapsulations in combating the resistance of different types of pathogens from a therapeutic point of view, as well as from a food point of view; finding strategies that increase the preservation of food sources but without harmfulness;
  5. The conclusions are consistent with the research objectives and open up new research opportunities;
  6. The bibliography is justifiable.

The authors wish to express their gratitude to the reviewer for the positive feedback provided on all sections of the article.

Round 2

Reviewer 2 Report

Comments and Suggestions for Authors

The revised version of the manuscript has answered all my recommendations. All modifications are suitable and have improved the quality of the manuscript thus making it suitable for publication.

Comments on the Quality of English Language

Minor English corrections are needed in the revised text (new added paragraphs) - e.g. lines 497-501